# Protective Effect of Follicle-Stimulating Hormone on DNA Damage of Chicken Follicular Granulosa Cells by Inhibiting CHK2/p53

**DOI:** 10.3390/cells11081291

**Published:** 2022-04-11

**Authors:** Shuo Zhou, An Zhao, Yangyang Wu, Tingting Bao, Yuling Mi, Caiqiao Zhang

**Affiliations:** College of Animal Sciences, Zhejiang University, Hangzhou 310058, China; shuozhou@zju.edu.cn (S.Z.); 21917037@zju.edu.cn (A.Z.); 22017122@zju.edu.cn (Y.W.); 21917093@zju.edu.cn (T.B.); yulingmi@zju.edu.cn (Y.M.)

**Keywords:** DNA damage, follicle-stimulating hormone, CHK2, p53, chicken

## Abstract

The increase in follicular atresia and the decrease in the fecundity of laying hens occur with the aging process. Therefore, the key measure for maintaining high laying performance is to alleviate follicular atresia in the aging poultry. Follicle-stimulating hormone (FSH), as an important pituitary hormone to promote follicle development and maturation, plays an important role in preventing reproductive aging in diverse animals. In this study, the physiological state of the prehierarchical small white follicles (SWFs) and atretic SWFs (ASWFs) were compared, followed by an exploration of the possible capacity of FSH to delay ASWFs’ progression in the hens. The results showed that the DNA damage within follicles increased with aging, along with Golgi complex disintegration, cell cycle arrest, increased apoptosis and autophagy in the ASWFs. Subsequently, the ACNU-induced follicular atresia model was established to evaluate the enhancing capacity of FSH on increasing cell proliferation and attenuating apoptosis in ASWFs. FSH inhibited DNA damage and promoted DNA repair by regulating the CHK2/p53 pathway. Furthermore, FSH inhibited CHK2/p53, thus, suppressing the disintegration of the Golgi complex, cell cycle arrest, and increased autophagy in the atretic follicles. Moreover, these effects from FSH treatment in ACNU-induced granulosa cells were similar to the treatment by a DNA repair agent AV-153. These results indicate that FSH protects aging-resulted DNA damage in granulosa cells by inhibiting CHK2/p53 in chicken prehierarchical follicles.

## 1. Introduction

The egg production and ovarian function of laying hens decreases significantly at the age of 80 weeks [1,2]. The increased atresia experienced by follicles was attributed to one of the main reasons for the decreasing egg production in 580-day-old (D580) hens. A prehierarchical follicle usually consists of one oocyte and numerous somatic cells, including the granulosa cells (GCs) and theca cells (TCs). The changes in GCs are especially obvious during follicular atresia. In the atresia follicles (AFs), cell apoptosis is observed earlier in GCs than in oocytes and TCs [3,4]. Compared with 280-day-old (D280) hens with peak laying performance, the number of prehierarchical follicles decreases rapidly and the number of AFs increases [5]. Therefore, it is important to prevent or postpone follicular atresia to maintain competitive laying performance.

The decision of apoptosis or survival of GCs is reported to be closely related to multiple hormones. Administration of a gonadotropin follicle-stimulating hormone (FSH) stimulates the secretion of steroid hormones and proliferation of GCs via the AKT pathway to promote cell survival [6,7]. FSH stimulates the production of intracellular cyclic adenosine monophosphate (cAMP) and activation of protein kinase A signaling pathway by binding with its receptor (FSHR) [8]. FSH induces increased steroidogenesis of the target cells; for example, FSH promotes the secretion of P_4_ from GCs [8,9] and the secretion of estradiol and prostaglandin from TCs of small yellow follicles [10]. FSH also promotes the proliferation of ovarian somatic cells and affects the secretion of diverse growth factors. In addition, FSH inhibits the expression of connective tissue growth factors in GCs of laying hens before ovulation. On the contrary, FSH promotes the expression of connective tissue growth factors in follicles after ovulation [11]. However, the function of FSH in the regulation of follicular atresia is still unclear in aging hens.

During the activation and development of follicles, a large number of the primordial follicles remain arrested in the first meiotic prophase. In this period, follicles are easy to subject to various endogenous and exogenous insults, resulting in DNA damage (single- and double-strand breaks). It is well known that DNA damage triggers DNA repair activation, cell–cycle arrest, and transcriptional changes [12,13,14]. Meanwhile, DNA damage is caused by excessive reactive oxygen species (ROS) during the aging process. Excessive ROS production causes DNA damage and activated/phosphorylated p53, which leads to aging [15]. Different from proteins, RNA, and lipids, DNA cannot be replaced after damage and must be repaired. When the damage is not repaired or abnormally repaired, cells may undergo apoptosis or necrosis.

DNA repair system and cell cycle checkpoint control maintain genomic integrity after DNA damage [16]. This damage leads to a predisposition to cancer and aging [17]. As a protein kinase, cell cycle checkpoint kinase 2 (CHK2) is a congener of Saccharomyces cerevisiae Rad53 and Schizosaccharomyces pombe Cds1. Importantly, CHK2 is related to the p53 pathway and regulates cell cycle arrest [18,19,20]. After DNA damage, CHK2 activates and phosphorylates p53. Various forms of cell stress induce a significant increase in p53 protein after translation [21,22]. Therefore, the normal function of CHK2 and its downstream p53 is crucial for DNA damage that may be elicited by aging.

Here, we studied the effect of DNA damage on the follicle atresia process, such as cell cycle arrest, Golgi complex disintegration, cell apoptosis, and autophagy. Furthermore, we investigated the attenuating effect of FSH on DNA damage in aging chicken prehierarchical follicles. Our findings may provide new understandings on the follicle atresia process and the effects of FSH involved in resistance to follicle atresia and ovarian aging in aged laying chickens.

## 2. Materials and Methods

### 2.1. Animals and Ethics

All hen procedures were performed in accordance with the *Guiding Principles for the Care and Use of Laboratory Animals of Zhejiang University* (ZJU20170660). Hyline white hens (*Gallus domesticus*) were purchased from the local farm and raised in the campus animal house. SWFs and ASWFs were collected from D280 hens (more than 5 hens). Ovaries were obtained under sterile conditions and extra tissues were removed using fine tweezers and scalpels. SWFs and ASWFs were used for subsequent H&E staining, β-galactosidase staining, immunofluorescence staining and Western blotting analysis. The experimental protocols were approved by the *Committee on the Ethics of Animal Experiments of Zhejiang University*.

### 2.2. Culture of Follicles and Treatment of Chemicals

The SWFs and ASWFs from high-laying (aged 280 ± 20 days) hens were transferred to the complete medium of DMEM high glucose (Hyclone, Tauranga, New Zealand) supplemented with 5% fetal calf serum (FCS; Hyclone, UT, USA). The follicles were cultured in 48-well culture plates (Corning Inc., Corning, NY, USA) at 38.5 °C and 5% CO_2_ atmosphere for 72 h. ASWFs were treated with FSH (0.1 IU/mL) for 72 h [9].

### 2.3. Cell Culture and Treatments

Small yellow follicles (SYFs, 6–8 mm) were removed from laying hens (aged 280 days) and placed in M199 medium (Hyclone, Tauranga, New Zealand). The granulosa layers were separated and digested with 1 mg/mL collagenase. The dispersed GCs were filtrated through a 200 μm mesh and centrifuged at 1000 rpm for 5 min. Trypan blue exclusion test was used to estimate the cell number and survival rate (routinely > 90%). A cell density of 4 × 10^5^/well was seeded into the 6-well culture plates (Nunc, Roskilde, Denmark) in 200 μL/well M199 medium with 1×ITS mixture (10 mg/mL insulin, 5 mg/mL transferrin and 30 nM selenite) and 2 mM glutamine. The GCs were cultured at 38.5 °C and 5% CO_2_. After 24 h, GCs were treated with FSH (0.01 IU/mL) for 24 h and/or ACNU (50 μM). In some experiments, GCs were treated with AV-153 (5 mM).

### 2.4. Cell Viability Assay

The viability of GCs was measured using Cell Counting Kit-8 (CCK-8; FD3788, Fudebio, Hangzhou, China). The experimental procedures were carried out following the manufacturer’s instructions. GCs were grown in 6-well plates to 90% confluency. CCK-8 assay reagent (10 μL) was added to each well containing 100 μL medium and incubated in the dark for 4 h at 38.5 °C. The formation of formazan was assessed by determining the optical density (OD) at 450 nm with a microplate spectrophotometer.

### 2.5. Morphological Observation

GCs were fixed in 4% paraformaldehyde solution for 2 h at 4°C. The primary antibodies used for the immunofluorescence were as follows: rabbit anti-FSHR (1:200, GB11275-1, Servicebio, Wuhan, China), rabbit anti-γH2AX (1:200, A11463, Abclonal, Wuhan, China), rabbit anti-CDKN1A (1:200, ER1906-07) and rabbit anti-CHK2 (1:200, ET1610-52, HUABIO, Hangzhou, China). Then, 20 μM 5-ethynyl-2′ -deoxyuridine (EdU, C0071S, Beyotime Biotechnology, Shanghai, China) was added to the cultured GCs for 2 h. Cells were counterstained with 4′,6-diamidino-2-phenylindole (DAPI, C1006, Beyotime Biotechnology, Shanghai, China). Mounted slides were visualized using an Olympus microscope (IX81, Tokyo, Japan).

### 2.6. Western Blot Analysis

The cultured cells were homogenized and quantified for the total protein using a BCA Protein Assay Kit (Nanjing Jiancheng Bioengineering Institute, Nanjing, China). Proteins (22 μg) were separated on a 10% SDS–polyacrylamide gel electrophoresis and transferred onto a polyvinylidene difluoride (PVDF) membrane (0.22 μm, Millipore, Massachusetts, Bedford, MA, USA). The PVDF membrane was incubated with corresponding primary antibodies including rabbit anti-CHK2 (1:500, ET1610-52), rabbit anti-CCND1 (1:500, RE6025), anti-p-p53 (1:500, ET1609-13), anti-caspase 3 (1:500, ER1802-42), anti-p53 (1:500, ET1602-38), anti-EAAT3 (1:500, ET1706-55), anti-CDKN1A (1:500, ER1906-07), anti-Rad51 (1:500, ET1705-96), anti-Bax (1:500, EM1203), anti-Bcl2A1 (1:500, ET1610-20), anti-LC3B (1:500, ET1701-65), anti-GOLPH3 (1:500, ER1706-39, HUABIO, Hangzhou, China), anti-FSHR (1:500, GB11275, Servicebio, Wuhan, China), rabbit anti-γH2AX (1:1000, A11463), anti-MYO18A (1:500, A9015, Abclonal, Wuhan, China) and anti-Atg7 (1:500, T57051, Abmart, Shanghai, China). Blots were washed three times in PBS with Tween and visualized using FDbio-Femto ECL Substrate Kit (FD8030, Fudebio, Hangzhou, China). For protein quantification, a Gel-Pro Analyzer (Media Cybernetics, Houston, TX, USA) was used to quantify and analyze images with β-actin as the internal control.

### 2.7. Transmission Electron Microscopy (TEM)

GCs were fixed in 2.5% glutaraldehyde for 24 h at 4 °C and dehydrated in ethyl alcohol and acetone. Then GCs were embedded in LX 112 epoxy resin. Sections with 70–90 nm thickness were cut with an ultramicrotome (Leica Microsystems GmbH, Wetzlar, Germany) and mounted on formvar-coated copper grids. The samples were observed and photographed with a Tecnai G2 Spirit (FEI Company, Hillsboro, OR, USA) after staining.

### 2.8. Transfection of CHK2 siRNAs, RNA Extraction and qRT-PCR

At 70% confluence of the cultured GCs, the cells were transfected for 24 h with either a SMART pool of small interfering RNA (siRNA) specific for CHK2 or non-silencing control (GenePharma Co., Ltd., Shanghai, China) using Lipofectamine 2000 (Meilun Bio, Dalian, China) in accordance with the manufacturer’s instructions. After 24 h, the transfection mixtures were replaced by the regular medium. The antisense sequences of primers for siRNAs are listed in Table 1.

Total RNA was extracted from cells using a Trizol reagent (Invitrogen Co., Carlsbad, CA, USA). The cDNA was generated with 2 μg total RNA by using a SuperScript First-Strand Synthesis System (Fermantas, Glen Burnie, MD, USA) based on the manufacturer’s protocol. Quantitative real-time PCR (qRT-PCR) was used to assess the expression of *FSHR*, *CHK2*, *CCND1*, *CDKN1A*, *γH2AX*, *Rad51*, *BCL2A1*, *BAX*, *Caspase3*, *LC3B*, *Atg7*, *MYO18A* and *GOLPH3*. The sequences of the primers for PCR analysis are listed in Table 2.

### 2.9. Immunocoprecipitation

Immunocoprecipitation was performed according to the instruction of Protein A/G immunoprecipitation magnetic beads (Bimake, Houston, TX, USA). The anti-CHK2 antibody was incubated for 2 h at room temperature for antibody immunobilization. After centrifugation of cell lysates, the supernatants were immunoprecipitated with a coupling antibody. After another centrifugation of immunoprecipitated supernatants, the protein complex was washed three times with RIPA lysis buffer. Immunoprecipitates were incubated with elution buffer for 5 min at room temperature.

### 2.10. Statistical Analysis

All experiments were repeated three times. Data were analyzed by one-way ANOVA with post hoc Dunnett’s test and independent samples t-test using the GraphPad Prism7 software (San Diego, CA, USA). *p* < 0.05 was statistically significant.

## 3. Results

### 3.1. Comparison of FSHR and DNA Damage between SWFs and ASWFs

As shown in Figure 1A, FSHR was predominantly expressed in the granulosa layer of both SWFs and ASWFs, besides the vitelline membrane, but weakly expressed in the ASWFs. The results of the β-galactosidase assay showed that the formation of aging cells was remarkably enhanced in the GCs of ASWFs (Figure 1B), as compared with that in SWFs. Meanwhile, γH2AX (a marker of DNA damage), CDKN1A (a cell cycle-related protein), CHK2 (a DNA damage response checkpoint protein) and p-CHK2 were significantly increased in ASWFs (Figure 1C–F). Meanwhile, the Golgi apparatus disintegrated in AFs, as observed by Transmission Electron Microscopy (TEM) (Figure 1G). Notably, the upregulation of CHK2, CDKN1A, p-p53/p53 and CCND1 in ASWFs further confirmed that DNA damage was increased during aging (Figure 1H).

### 3.2. Alleviating Effect of FSH on GC DNA Damage via Inhibiting CHK2

GCs were treated with different doses of ACNU (25, 50, 100 μM). The results showed that ACNU treatment significantly inhibited the viability of GCs. Treatment with 50 μM ACNU markedly reduced the viability of GCs by 12.94% and higher ACNU (100 μM) further decreased cell viability by 19.17% (Appendix A). Treatment with 0.01 and 0.1 IU/mL FSH remarkably increased (37.55% and 52.00%) the cell viability (Appendix A). The effect of ACNU on the proliferation of GCs was detected by EdU incorporation. The decreased proliferation of GCs caused by ACNU (50 μM) was significantly increased in the presence of FSH (0.01 IU/mL) (Appendix A).

The cultured GCs were treated with 50 μM ACNU and 0.01 IU/mL FSH, alone or in combination. As shown in Figure 2A, the decreased viability of GCs that was caused by 50 μM ACNU was significantly enhanced in the presence of 0.01 IU/mL FSH. After treatment with FSH for 24 h, the cultured GCs showed a marked reduction in the expression of CHK2 and γH2AX, after being treated with ACNU. Correspondingly, the Western blot experiment also showed that the CDKN1A expression and Rad51 degradation after ACNU exposure were significantly inhibited by FSH. Meanwhile, RT-qPCR determination showed the same results (Figure 2B). Moreover, the immunofluorescence assay showed that FSH reduced the elevated the expression of CHK2, caused by ACNU in GCs (Figure 2C).

### 3.3. Alleviating Effect of FSH on Cell Cycle Arrest

Cell cycle arrest was caused by ACNU-induced DNA damage. The results showed that most of the GCs were in the S phase and G1 phase after ACNU treatment. However, simultaneous treatment of FSH effectively alleviated the cell cycle arrest caused by DNA damage (Figure 3A–E). Correspondingly, the Western blot experiment also showed that the decrease in CCND1 expression after ACNU treatment was remarkably elevated by FSH treatment (Figure 3F). Furthermore, the immunofluorescence assay showed that, compared with ACNU treatment, FSH treatment decreased the expression of PUMA by 24.78% (Figure 3G).

### 3.4. Decrease of Apoptosis after FSH Treatment

The increased apoptosis of GCs caused by ACNU was significantly decreased in the presence of FSH (Figure 4A). The results of the Western blot analysis showed that treatment of FSH reduced the expression of Bax and Caspase 3 and increased the expression of Bcl2A1 in the ACNU-treated cells (Figure 4B).

### 3.5. The alleviating Effect of FSH on Golgi Complex and Autophagy

Western blot and RT-qPCR experiments showed that the LC3B and Atg7 expression after ACNU exposure were significantly inhibited by FSH treatment (Figure 5A). Because of the important role of GOLPH3/MYO18A in the maintenance of Golgi complex morphology (Farber-Katz et al., 2014; Buschman et al., 2015), we tested whether GOLPH3 and MYO18A are required for DNA damage-induced dispersal of the Golgi complex. The results showed that DNA damage caused the decrease in Golgi matrix disintegration-related proteins/mRNA MYO18A and GOLPH3, which were alleviated by FSH treatment (Figure 5B).

### 3.6. Reduction in DNA Damage after Treatment with AV-153

Treatment with 5 μM and 10 μM AV-153 remarkably increased the viability of GCs (Figure 6A). The results of the Western blot showed that the p-CHK2/CHK2, CDKN1A, γH2AX and LC3B expression and Rad51 degradation after ACNU induction were significantly inhibited by FSH, similar to the effect of AV-153 treatment (Figure 6B).

### 3.7. Interaction of CHK2 with FoxK and FoxO3a

The co-immunoprecipitation assay confirmed the binding between CHK2, FOXK and FoxO3a proteins. The interaction between CHK2 and FOXK or FoxO3a was confirmed using a reciprocal coimmunoprecipitation assay (Figure 7A). Knockdown of CHK2 in GCs by each of the three siRNA oligos decreased the expression of CHK2 mRNA and protein (Figure 7B). We depleted CHK2 in GCs and then examined the levels of DNA damage, cell cycle and autophagy by assessing the p53, CCND1, CDKN1A, Rad51, LC3B and Atg7 protein levels. As shown in Figure 8C, depletion of CHK2 resulted in a significant increase in DNA damage, as demonstrated by a reduction in Rad51 protein level and an increase in CDKN1A level. Meanwhile, depletion of CHK2 resulted in a significant increase in autophagy and cell cycle arrest (Figure 7C).

### 3.8. Decreased DNA Damage in ASWF GCs and ASWF after FSH Treatment

In addition to the exploration of the effect of FSH on follicle aging, GCs were isolated from ASWFs from D280 hens and treated with FSH (0.01 IU/mL) in culture. Both the cell viability and DNA damage were relieved in GCs from ASWFs (Figure 8A,B). Next, ASWFs were treated with FSH (0.1 IU/mL) for 72 h in culture. The FSHR expression was significantly elevated in ASWFs after FSH treatment (Figure 8C). Meanwhile, treatment of FSH increased cell proliferation and DNA repair in ASWFs. Western blot experiments showed that the PCNA and Rad51 were increased after treatment of FSH in ASWFs. The expression of caspase 3 was decreased in FSH-treated ASWFs. Moreover, treatment with FSH significantly increased the expression of FSHR and decreased the expression of CHK2 and γH2AX (Figure 8D).

## 4. Discussion

After 80 weeks, laying performance declines sharply due to the aging ovaries of the layer. The aging ovary is featured by retarded follicle development and increased follicle atresia, leading to less and less follicle maturation and ovulation. It is reported that the number of atretic follicles increased during the laying hens’ aging process [23]. Follicle development is under the complex coordination of numerous hormones, cytokines and growth factors, such as reproductive axis hormones (GnRH, FSH and sex steroids), transforming growth factor (TGF-β) and fibroblast growth factor (bFGF), etc. [9,24,25]. Pituitary FSH is the predominant tropic hormone for gamete production. After binding with its receptor FSHR, FSH is capable of promoting the progressive development of prehierarchical follicles to hierarchical follicles in laying hens. In addition, FSH increases the levels of progesterone, E_2_ and androgen in plasma [26]. It has been reported that the levels of estradiol (E_2_) and anti-Müllerian hormone (AMH) increased after the injection of FSH into the low-birth-laying hens [9]. In this study, the effect of FSH on atresia of the prehierarchical SWFs was investigated, since follicles at this stage are critical for further transition into a higher developmental stage. The results showed that the DNA damage was increased and the expression of *FSHR* mRNA was reduced in the atretic follicles. Meanwhile, FSH delayed follicular atresia by decreasing DNA damage in the follicular granular layer.

Several products of tumor suppressor genes are at the core of aging processes, such as Rb, p16INK4a, p53 and p14ARF. The p53 is an important tumor suppressor gene. It was reported that the positive rate of invasive disease-free survival in breast cancer tissue showed a poor prognosis when the p53 protein was positive [27]. The p53 protein is a transcription factor that controls cell cycle initiation. The mutation of the *p53* gene results in overexpression and dysfunction of p53 protein, which decreases its roles in cell growth, apoptosis and DNA repair [28]. Burns et al. reported that CPEB binds to the 3 ‘-untranslated regions of p53 and Gld4. The translation control region of *p53* mRNA causes cell aging [29]. Our results showed that the p53 signal pathway was augmented in ASWFs and the ACNU-treated GCs. These results suggest that the p53 pathway plays an important role in the aging process of GCs from laying hens. Meanwhile, DNA damage was closely related to the p53 signaling pathway.

Previous studies showed that follicular atresia is related to DNA damage. DNA double-strand breaks (DSBs) are the most serious form of damage in eukaryotic cells, which damage the two strands of DNA double helix [30]. DSBs are usually caused by particle radiation, genotoxic chemicals, radiomimetic drugs and cancer chemotherapy. If DSBs are repaired incorrectly or not in time, it will lead to genome instability, even changes in gene dose. Chromosomal abnormalities cause tumors or other aging-related diseases [31]. If cells do not completely repair before DNA replication, permanent irreversible chromosome damage will arise. In this case, cell cycle checkpoint and DNA damage signal make the cell exit the cell cycle irreversibly. It is reported that DNA damage and abnormal telomeres activated the p53-dependent DNA damage pathway and ATM/ATR. Meanwhile, DNA damage usually induces its downstream target cyclin-dependent kinase inhibitor CDKN1A/p21WAF1/CIP1 (p21), resulting in G1 cell cycle arrest and accumulation of p53 in cells and induces apoptosis by activating p53, which mediates the transcriptional activation of pro-apoptotic proteins [32,33,34]. The effect of DNA damage on the Golgi complex has been largely ignored. Even a minimal dose of DNA damage agent induces Golgi complex dispersal [35]. Time-lapse microscopic observation of cell surface and Golgi complex dispersal for several weeks showed that Golgi complex reaction is not likely related to cell apoptosis [36]. The p53 also plays an important role in regulating autophagy [37]. The p53 enhances the expression of Sestrin1 and Sestrin2 to activate AMPK, thereby preventing mTOR in the nucleus [38,39]. However, p53-induced proapoptotic factors, such as p53-induced BH3-only protein (PUMA), also activate autophagy in the cytoplasm [40]. Follicular atresia was accompanied by apparent DNA damage (Figure 1). In this study, DNA damage markers and p53 phosphorylation increased in the AFs. Furthermore, DNA damage elicited apoptosis, cell cycle arrest, Golgi matrix disintegration, and autophagy of GCs from laying hens (Figure 3, Figure 4, Figure 5 and Figure 6).

Follicular development includes oocyte growth, follicular cell proliferation, differentiation, and apoptosis. However, more than 99% of follicles will become atretic and the fundamental reason for follicular atresia is the apoptosis of GCs. It has been reported that downstream cytokines related to the cell cycle were activated after CHK1/2 activation. On the contrary, the preantral follicles failed to grow or even degenerated after inhibition of CHK1/2 [32]. CHK1/2 is necessary for the development of ovarian follicles. CHK2 is ubiquitous in mammals, as an important signal transduction protein in DNA damage response. It has been found to participate in a variety of physiological or pathological activities in humans. CHK2-deficient cells showed G1/S arrest of the cell cycle [32]. In the process of human EBV infection, CHK2 directly interacts with nuclear antigen 3C (EBNA3C) to inhibit the G2/M phase arrest of the cell cycle, thus, preventing the process of virus infection [41]. By downregulating the cell cycle checkpoint proteins cyclin B1 and CHK2, colon cancer Caco-2 cells showed an anti-proliferative effect. It has also been reported that the CHK2 protein plays an important role in the occurrence of prostate cancer, ovarian cancer, and other tumors [41]. Our results showed that CHK2 was activated in the ASWFs and FSH decreased the elevated CHK2 that was induced by DNA damage, thereby relieving follicular atresia. Therefore, FSH maintains GC survival against cell cycle arrest, Golgi complex dispersal, apoptosis, and autophagy through the inhibition of DNA damage (see Graphical abstract).

## 5. Conclusions

In conclusion, increased DNA damage and decreased DNA repair occurred in the ASWFs of the laying chicken. Treatment of FSH significantly restored the increased DNA damage of ACNU-elicited cultured GCs and naturally atretic prehierarchical follicles. FSH was able to increase the survival of GCs from oxidative stress-induced DNA damage by suppressing DNA damage through the CHK2/p53 pathway. Therefore, targeting CHK2/p53 signaling might provide a valuable measure of retarding follicular atresia or ovarian aging in laying chickens.

## Figures and Tables

**Figure 1 cells-11-01291-f001:**
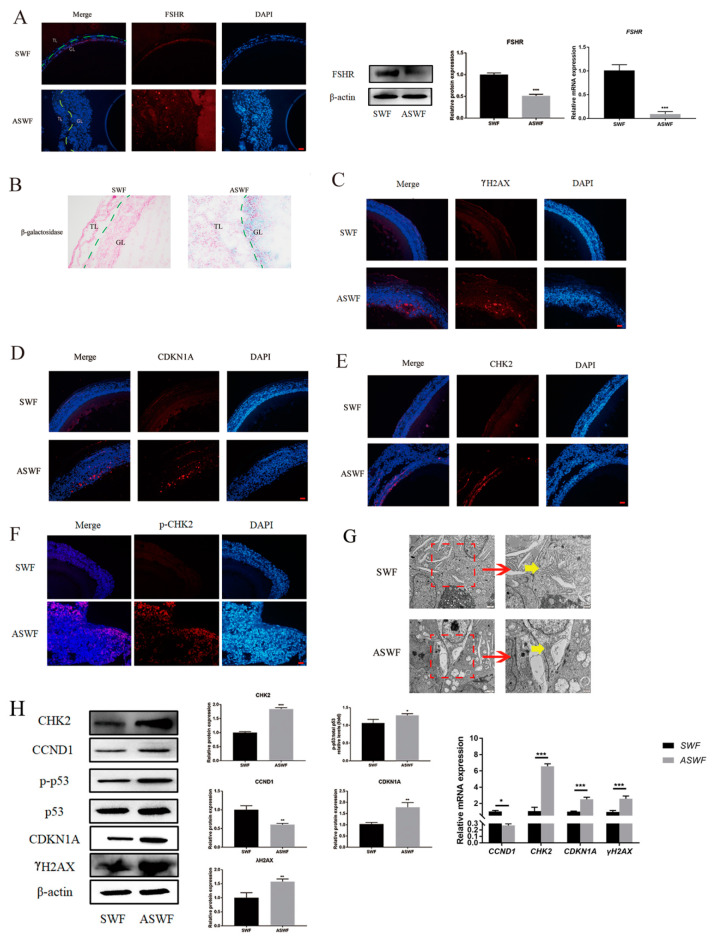
Morphological changes between SWFs and ASWFs. (**A**) The expression of FSHR in SWFs or ASWFs. Scale bar: 20 µm. GL: Granulosa layer; TL: Theca layer. (**B**) β-galactosidase staining of SWFs or ASWFs. Scale bar: 50 µm. (**C**) The expression of γH2AX in SWFs or ASWFs. Scale bar: 20 µm. (**D**) The expression of CDKN1A in SWFs or ASWFs. Scale bar: 20 µm. (**E**) The expression of CHK2 in SWFs or ASWFs. Scale bar: 20 µm. (**F**) The expression of p-CHK2 in SWFs or ASWFs. Scale bar: 20 µm. (**G**) The ultrastructure of SWFs or ASWFs in hens aged 280 days. Scale bar: 1 µm (left) or 500 nm (right). (**H**) Relative expression of proteins related to DNA damage in the follicles. * *p* < 0.05, ** *p* < 0.01, *** *p* < 0.001.

**Figure 2 cells-11-01291-f002:**
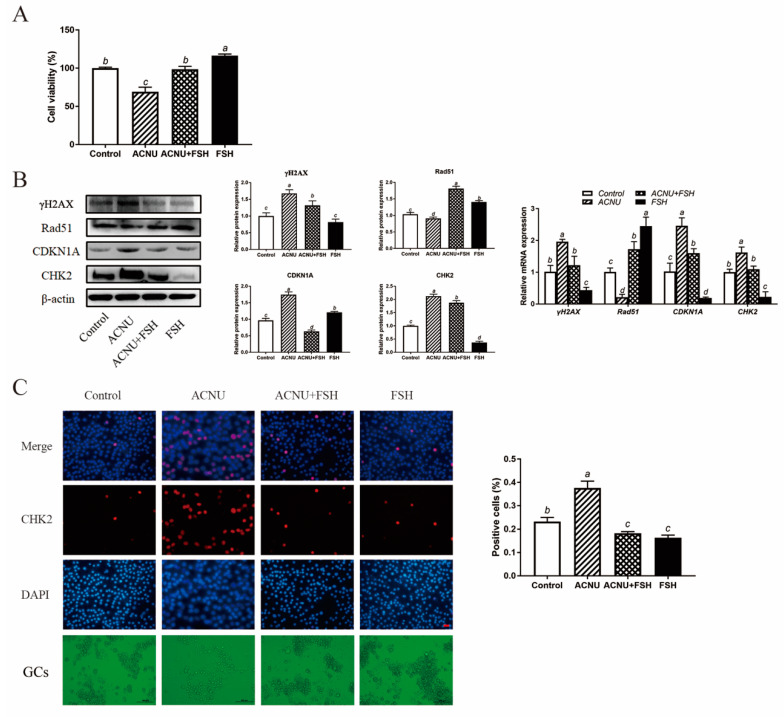
Effect of FSH on DNA damage of cultured GCs via CHK2 activation. GCs were treated with 0.01 IU/mL FSH and/or 50 μM ACNU for 24 h. (**A**) Changes in cell viability using the CCK-8 assay. Values are the means ± SEM (*n* = 3). Bars with different superscripts are statistically different (*p* < 0.05). (**B**) Relative expression of proteins and mRNA related to DNA damage in the follicles. Bars with different superscripts are statistically different (*p* < 0.05). (**C**) The expression of CHK2 in GCs after treatment of ACNU and/or FSH. Scale bar: 20 µm.

**Figure 3 cells-11-01291-f003:**
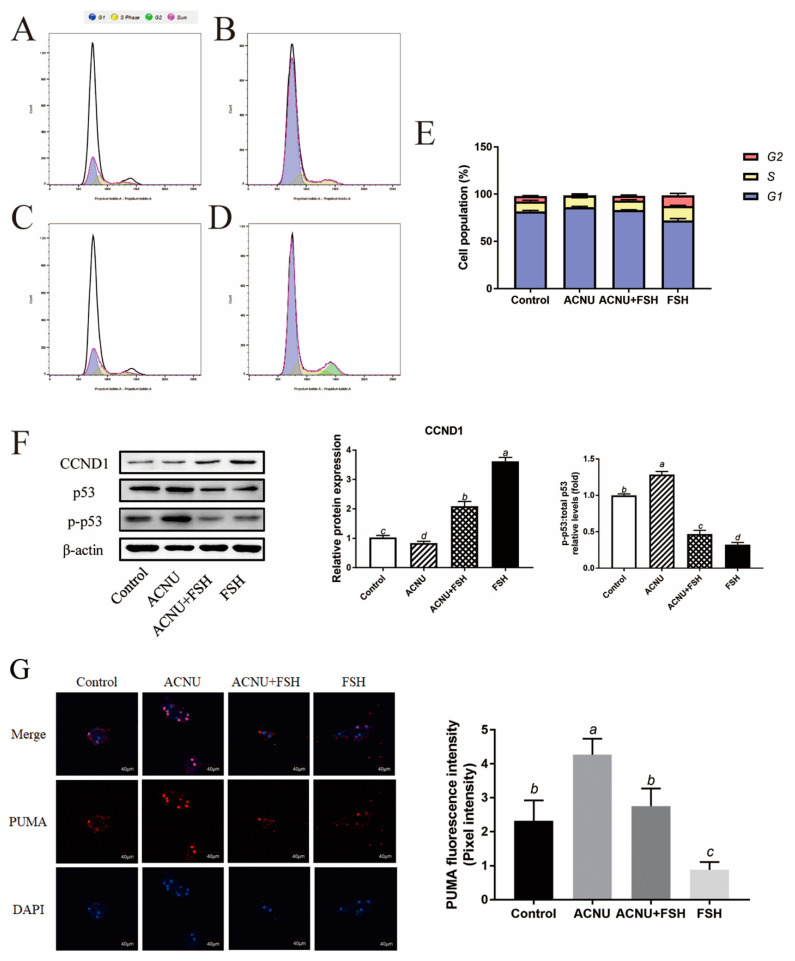
Changes in cell cycle after FSH treatment. (**A**–**E**) Cell apoptosis using the Flow cytometry assay. Values are the means ± SEM (*n* = 3). (**A**) Control; (**B**) ACUN treatment; (**C**) ACNU and FSH treatment in combination; (**D**) FSH treatment; (**E**) The analysis of cell cycle. (**F**) Relative expression of proteins related to apoptosis caused by DNA damage in GCs treated with ACNU and/or FSH. (**G**) The expression of PUMA in GCs after the treatment of ACNU and/or FSH. Scale bar: 40 µm. Bars with different superscripts are statistically different (*p* < 0.05).

**Figure 4 cells-11-01291-f004:**
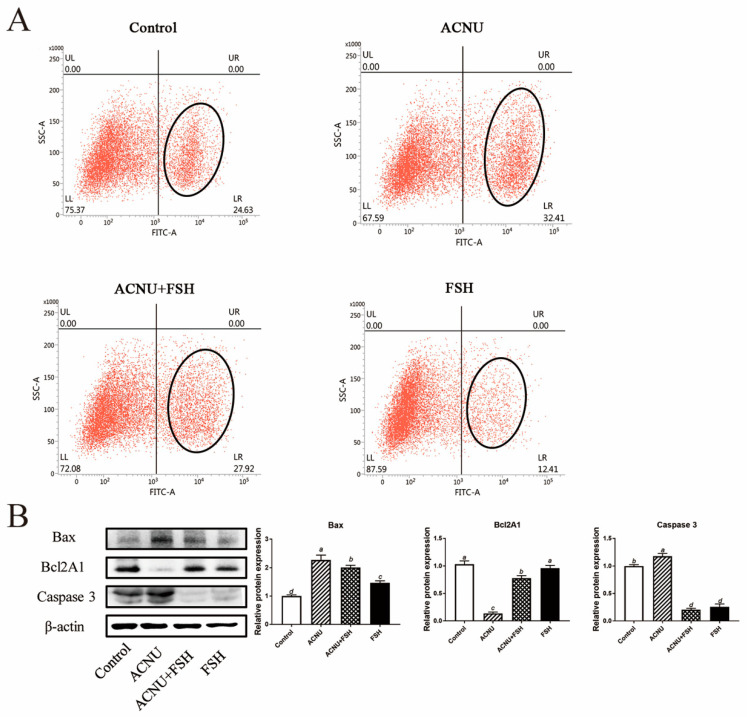
Changes in cell apoptosis after FSH treatment. (**A**) Cell apoptosis using the Flow cytometry assay. Values are the means ± SEM (*n* = 3). (**B**) Relative expression of proteins related to apoptosis caused by DNA damage in GCs treated with ACNU and/or FSH. Bars with different superscripts are statistically different (*p* < 0.05).

**Figure 5 cells-11-01291-f005:**
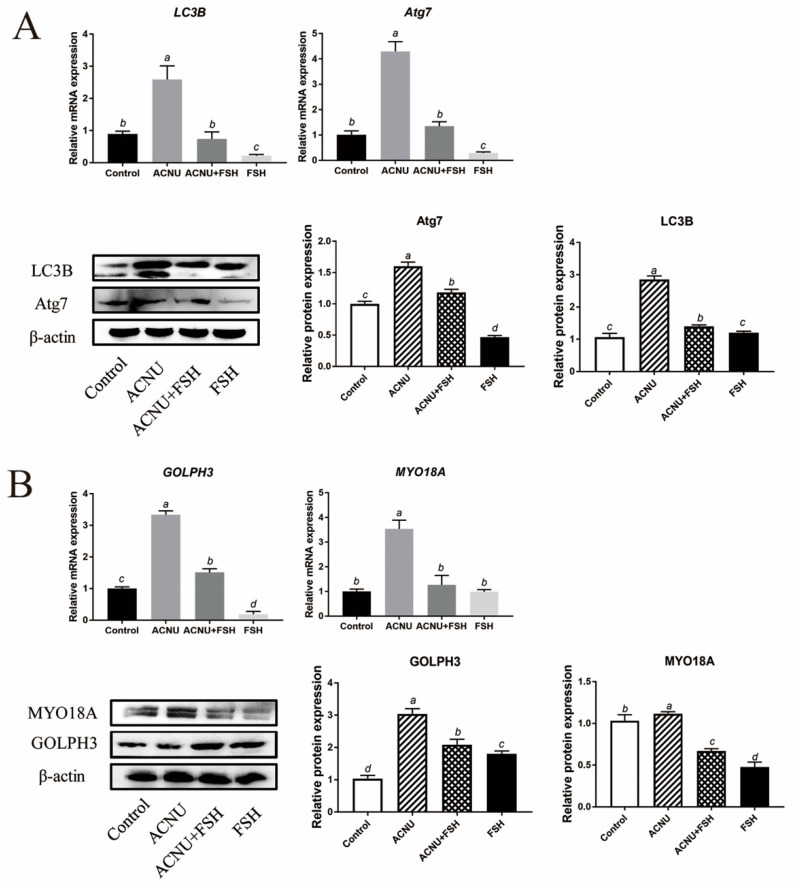
Changes in Golgi complex and autophagy after FSH treatment. (**A**) Relative expression of proteins and mRNA related to autophagy in GCs treated with ACNU and/or FSH. Bars with different superscripts are statistically different (*p* < 0.05). (**B**) Relative expression of proteins and mRNA related to Golgi complex morphology caused by DNA damage in GCs treated with ACNU and/or FSH. Bars with different superscripts are statistically different (*p* < 0.05).

**Figure 6 cells-11-01291-f006:**
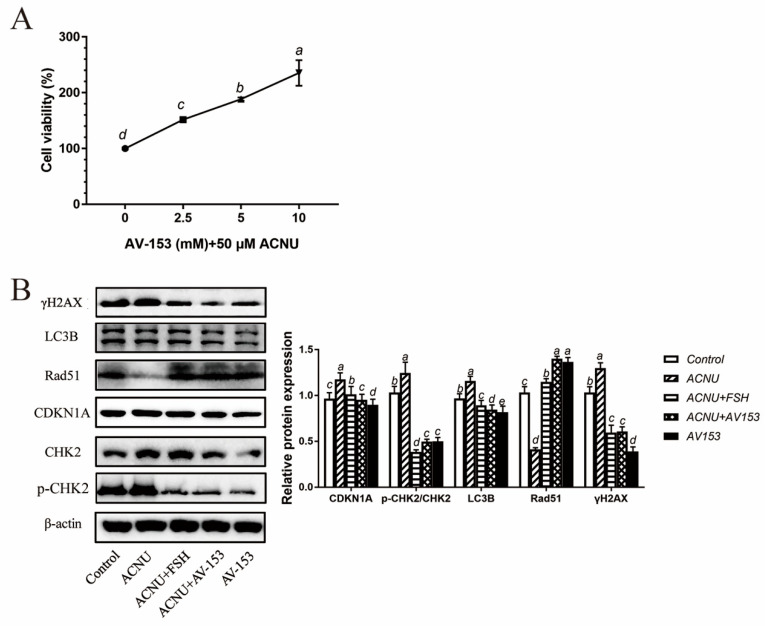
Reduced DNA damage after treatment of FSH and AV-153 in GCs. GCs were treated with 50 μM ACNU, AV-153 (5 mM) or FSH (0.01 IU/mL) alone or in combination. (**A**) Cell viability determined with the CCK-8 assay in D280 hen GCs treated by AV-153 in culture. Values are the means ± SEM (*n* = 3). Bars with different superscripts are statistically different (*p* < 0.05). (**B**) Relative expression of proteins related to DNA damage in GCs treated with ACNU, AV-153 or FSH. Bars with different superscripts are statistically different (*p* < 0.05).

**Figure 7 cells-11-01291-f007:**
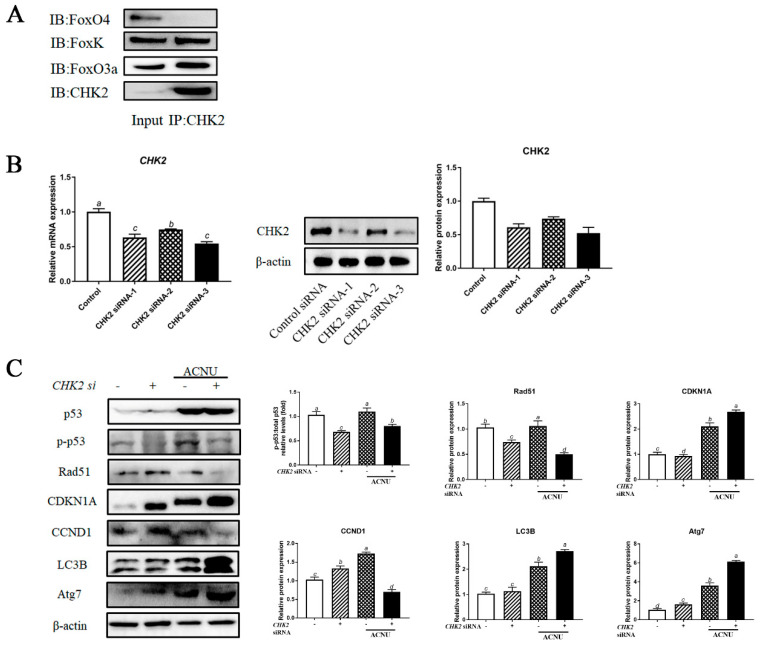
The interactions of CHK2 with FoxK or FoxO3a. (**A**) The cell lysates were processed for coimmunoprecipitation with anti-CHK2, followed by probing with anti-FoxO3a, FoxK and FoxO4. IP, immunoprecipitation. (**B**) GCs transfected with CHK2 siRNA or scrambled control siRNA for 24 h/48 h were cultured for 24 h and then the cells were rinsed with PBS. The expression of CHK2 was determined by qPCR and the expression of CHK2 protein was determined by Western blot. Bars with different superscripts are statistically different (*p* < 0.05). (**C**) GCs transfected with CHK2 siRNA for 24 h were cultured in media containing ACNU. 24 h later, the cells were rinsed with PBS and the expressions of p53, p-p53, Rad51, CDKN1A, CCND1, LC3B and Atg7 were determined by Western blot. Bars with different superscripts are statistically different (*p* < 0.05).

**Figure 8 cells-11-01291-f008:**
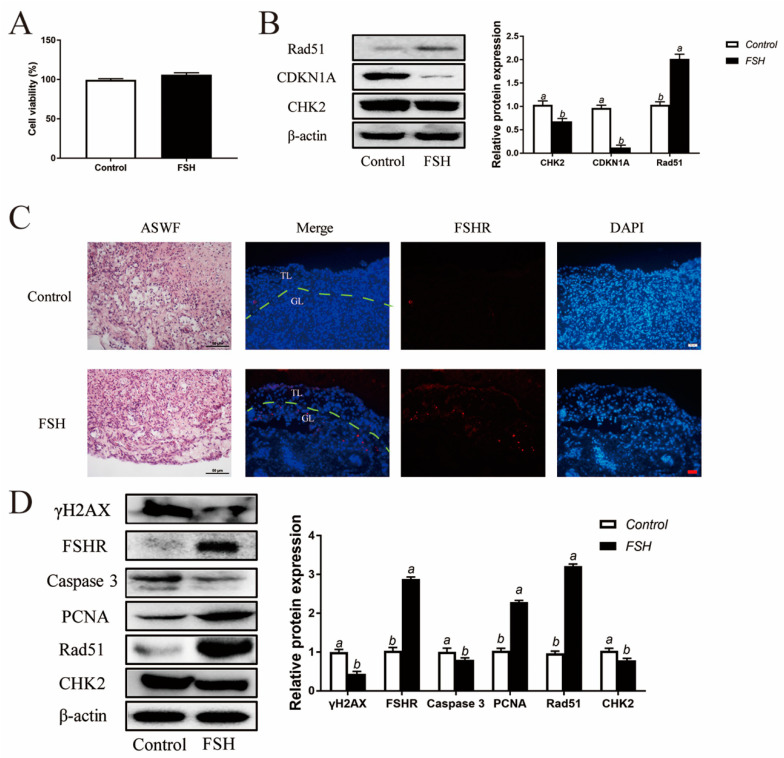
Reduced DNA damage after FSH treatment in ASWFs. (**A**) GCs from ASWFs were treated with 0.01 IU/mL FSH in culture. Cell viability using the CCK-8 assay in GCs from ASWFs treated with FSH in culture. Values are the means ± SEM (*n* = 3). Bars with different superscripts are statistically different (*p* < 0.05). (**B**) Relative expression of proteins related to DNA damage in GCs from ASWFs treated with FSH. Bars with different superscripts are statistically different (*p* < 0.05). (**C**) ASWFs were treated with 0.1 IU/mL FSH for 72 h. Representative morphology and the expression of FSHR of ASWFs treated with FSH in culture. Scale bar in H&E staining: 50 µm; Scale bar in IF staining: 20 µm. GL: Granulosa layer; TL: Theca layer. (**D**) Relative expression of γH2AX, FSHR, caspase3, PCNA, Rad51 and CHK2 proteins in cultured ASWFs treated with FSH. Bars with different superscripts are statistically different (*p* < 0.05).

**Table 1 cells-11-01291-t001:** Antisense primers for siRNA.

Gene Name	Sense (5′-3′)	Antisense (5′-3′)
CHK2-gallus-301	GUUCCCACCUUAAUUACAUTT	AUGUAAUUAAGGUGGGAACTT
CHK2-gallus-1014	GUCCUCAUGAACAAGCAGUTT	ACUGCUUGUUCAUGAGGACTT
CHK2-gallus-219	GCCUUUAUGUUCCUCCUCATT	UGAGGAGGAACAUAAAGGCTT
NC	UCUCCGAACGUGUCACGUTT	ACGUGACACGUUCGGAGAATT

**Table 2 cells-11-01291-t002:** Primers for PCR analysis.

Genes	Accession No.	Primer Sequence (5′–3′)
*FSHR*	XM_046913182.1	GAGCCACCACGCTGAAGACATC ACCACTCCGCAGTCCTGTTACC
*CHK2*	NM_001397669.1	GATCTCACGGTGGATGACCAGTTG CCACAGGCACCACTTCCCAAAG
*CCND1*	XM_046917934.1	CCGAAGGTTGTGTTCCAGTGAGAG CGTGTGTTGGCACCAAAGGATTTC
*CDKN1A*	XM_046932838.1	GTGACTGCTGCTGCTGAGGATG AACTACAGACTCGGCATTGCTTCG
*γH2AX*	BM488821.1	AACAAGAAGACGCGCATCATCCC GTAGCACGGCCTGAATGTTGGG
*Rad51*	NM_205173.2	CTATGTGCTGCTGGTGGCTGATAC CACTGACACTGGAGGACTGTGATTG
*BCL2A1*	NM_204866.2	CCTGATTAAGAGCAGCACACTGGTC GGTCCGAGATGTGATTCCTGAAGC
*BAX*	XM_040693909.2	ACTCTGCTGCTGCTCTCCTCTC CCGCTCTCTGCCTTCTCAATGATG
*Caspase3*	NM_204725.2	ATTGAAGCAGACAGTGGACCAGATG TGCGTTCCTCCAGGAGTAGTAGC
*LC3B*	AW240014.1	TCCAACCTTGACAGACACAACCTTC GGGCGATGGCACTTCTACTGATTC
*Atg7*	NM_001396468.1	CGCTGGAGCCGTGAACTACAATC GGCACTACTGAAGGGAGCAAACTG
*MYO18A*	XM_046930299.1	ATGCTGGACCACCTGAAGAACAAC ATGGACTTGCGGGCTTTGACAG
*GOLPH3*	XM_046936332.1	AGGCTAACAATGGCGGTAAGTTCAG TGGATCTCAGCAGAACAAAGCAGTC
*β-actin*	NM_205518	ACACCCACACCCCTGTGATGAA TGCTGCTGACACCTTCACCATTC

## Data Availability

All data analyzed during this study are included in the published article.

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
