# Peer review of "Protective Effect of Follicle-Stimulating Hormone on DNA Damage of Chicken Follicular Granulosa Cells by Inhibiting CHK2/p53"

_cells, 2022, doi:10.3390/cells11081291_

Round 1

Reviewer 1 Report

At the moment, the revised manuscript entitled “Protective effect of follicle-stimulation hormone on DNA damage of chicken follicular granulosa cells by inhibiting CHK2/p53” cannot find its place in Cells, a more accurate presentation of data is still required.

Major concerns:

Fig. 1 - The authors added gammaH2Ax staining to prove the existence of DNA damage in ASWF, yet this finding should be reinforced by IF assay for pho-CHK2 of the sections. Indeed, western blotting analysis of CHK2 presented in FIG1 panel G is not so clear: they showed only the total level of the CHK2. The authors speculated that DNA damage in ASWF leads to cell cycle arrest, yet to prove this, they should give present more experimental evidence: i.e. performing IF assay for cleaved Parp to test whether cells H2AX positive are still viable.

Fig.2B the quality of the WB is very low, the staining of beta-actin is saturated, this makes very difficult to quantify the relative protein expression of different markers.

Fig. 3F the quality of the WB for p-p53 is very low, therefore the authors could add the staining of some apopotic markers, (i.e. PUMA, NOXA?) direct targets of p-p53.

Fig.5B the quality of the WB is very low (in particular the staining for GOLPH3, an MYO18), and again the staining of beta actin is saturated, making very difficult the quantification of the relative protein expression of different markers.

CHK2 interacts with FoxK and Fox3a The authors should introduce better, why they decided to perform such set of experiments. It is not clear to me why a significant increase in DNA damage was demonstrated by a reduction in RAD51 protein level. Again, the quality of the WB presented in Fig.7C is very low. Could the authors make a comment on the CDKN1A staining? It seems that CDKN1A is shifted following ACNU treatment.

Author Response

Response to Reviewer

Re: cells-1623767

Title: Protective effect of follicle-stimulating hormone on DNA damage of chicken follicular granulosa cells by inhibiting CHK2/p53

We appreciate the review’s comments for improving the quality of our manuscript. The revisions were highlighted in yellow in the manuscript. The detailed responses to each question were listed as follows:

Major concerns:

Fig. 1 - The authors added gammaH2Ax staining to prove the existence of DNA damage in ASWF, yet this finding should be reinforced by IF assay for pho-CHK2 of the sections.

Response: The IF assay for pho-CHK2 was added.

Indeed, western blotting analysis of CHK2 presented in Fig.1 panel G is not so clear: they showed only the total level of the CHK2. The authors speculated that DNA damage in ASWF leads to cell cycle arrest, yet to prove this, they should give present more experimental evidence: i.e. performing IF assay for cleaved Parp to test whether cells H2AX positive are still viable.

Response: It was revised. Compared with SWF, there are more DNA breaks and the weakness ability of DNA repair in ASWF via IF assay for cleaved Parp.

Fig.2B the quality of the WB is very low, the staining of beta-actin is saturated, this makes very difficult to quantify the relative protein expression of different markers.

Response: We re-examined CDKN1A protein by Western blot. Due to species limitation, many antibodies are not ideal for chicken.

Fig. 3F the quality of the WB for p-p53 is very low, therefore the authors could add the staining of some apopotic markers, (i.e. PUMA, NOXA?) direct targets of p-p53.

Response: We re-examined p-p53 protein by Western blot. We also added the IF staining for PUMA.

Fig.5B the quality of the WB is very low (in particular the staining for GOLPH3, an MYO18), and again the staining of beta actin is saturated, making very difficult the quantification of the relative protein expression of different markers.

Response: We re-examined GOLPH3 and MYO18 protein by Western blot. Due to species limitation, MYO18 antibodies was not ideal for chicken.

CHK2 interacts with FoxK and Fox3a. The authors should introduce better, why they decided to perform such set of experiments.

Response: CHK2 is located in the nucleus. FSH cannot function directly into the nucleus, but only by promoting Fox family members. RNA-seq showed that FoxO3a, FoxK and FoxO4 in GCs were significantly increased. We tested whether CHK2 co-immunoprecipitation with FOXO3a, FoxK and FoxO4. We speculated that FSH promoted the transcription of FOXO3a and FoxK into the nucleus and make them interact with CHK2.

It is not clear to me why a significant increase in DNA damage was demonstrated by a reduction in RAD51 protein level.

Response: Rad51 is an important component of DNA double strand break repair. DNA damage significantly increased, while DNA repair ability decreased, leading to apoptosis.

Again, the quality of the WB presented in Fig.7C is very low. Could the authors make a comment on the CDKN1A staining? It seems that CDKN1A is shifted following ACNU treatment.

Response: Due to species limitation, many antibodies are not ideal for chicken. The IF staining of CDKN1A was added.

Reviewer 2 Report

This paper provides a highly amount of experiments done to highlight the protective effect of Follicle-stimulating hormone on DNA damage. The broad number of assays performed are more than enough to reach the point on the effect of FSH. There are instead, several point that need to be improved in order to better communicate to the reader the results you obtained.

-Since the experiments put into perspective the role of FSH, there must be a separate Figure describing the differences in the expression of FSHR between SWF and ASWF. I propose to divide the Immunofluorescence assay picture showing the expression of FSHR and the Western blot analysis indicating the same result into a separate Figure as the start of the results chapter. This way the reader is presented with the first insight that brought you to the conclusion that the cells must be treated with FSH.

-The titles of the Results seem repetitive because some of them start the same, so the point of the upcoming experiment is confusing. Either rephrase the titles or include some of the results chapters into one. Examples: 3.1 Comparison of DNA damage

                            3.2 Effect of DNA damage on GCs via CHK2 activation

                            3.6 Reduced DNA damage after FSH treatment the same as AV-153 in GCs

                            3.8 Reduced DNA damage after FSH treatment in ASWF GCs and ASWF.

- Pictures are not indicated clearly. They are divided into Letters: A,B,C etc. but under the same letter there are 2 or more different assays/results demonstrated. Example: Figure S1C, Figure 3F, Figure 4B, Figure 6B etc.

-The exact concentration of the compound that the samples are treated need to be listed together. They are listed in separate parts of the chapter, which makes them very confusing. The reader needs to return to the start of the paragraph to put altogether the different conditions on the samples.

-The results need to be commented more. The difference between the treated samples with FSH and the untreated one can be easily distinguished by the graphs, but there is a lack of comments regarding these graphs. They are usually described as: “Significant” or “effectively”. These creates a confusion on the optimal amount of compound needed to obtain a positive result.

Minor Revision:

The English language in general lacks clarity. There is a significant number of phrases that do not bring the point intended by the writer. Examples:

Line[56-57]: This phrase is missing “is” between “damage” and “caused”.

Line[59-60]:  Instead of “don’t” it should be used “cannot”.

Line[89-90]: There is a “in” missing between “cultured” and 48-well culture plate.

Line[93]: “were” instead of “was”

Line[175-176]: This sentence is incorrect. The sentence should be: “Meanwhile, the Golgi apparatus disintegrates in atresia follicles as observed by Transmission Electron Microscopy (TEM).

Line[181]: “off” should be replaced with “between”.

Line[205-206]: This sentence should be rephrased in order to highlight that the treatment with FSH shows a marked reduction in the expression of CHK2. The way this sentence is composed, is misleading and gives the illusion that after the treatment  with ACNU it was obtained the wanted effect on CHK2.

Figure 4E is not present in the Figure .

Author Response

Response to Reviewer

Re: cells-1623767

Title: Protective effect of follicle-stimulating hormone on DNA damage of chicken follicular granulosa cells by inhibiting CHK2/p53

We appreciate the review’s comments for improving the quality of our manuscript. The revisions were highlighted in yellow in the manuscript. The detailed responses to each question were listed as follows:

-Since the experiments put into perspective the role of FSH, there must be a separate Figure describing the differences in the expression of FSHR between SWF and ASWF. I propose to divide the Immunofluorescence assay picture showing the expression of FSHR and the Western blot analysis indicating the same result into a separate Figure as the start of the results chapter. This way the reader is presented with the first insight that brought you to the conclusion that the cells must be treated with FSH.

Response: It was revised.

-The titles of the Results seem repetitive because some of them start the same, so the point of the upcoming experiment is confusing.

Response: They were rephrased.

- Pictures are not indicated clearly. They are divided into Letters: A,B,C etc. but under the same letter there are 2 or more different assays/results demonstrated. Example: Figure S1C, Figure 3F, Figure 4B, Figure 6B etc.

Response: Figure S1C showed that the alleviating effect of FSH on the decreased proliferation of GCs induced by ACNU via EdU incorporation.

Figure 3F showed that FSH alleviated cell cycle arrest by activating p53 signaling pathway.

Figure 4B: Bax, Caspase 3 and Bcl2A1 are all associated with apoptosis.

Figure 6B showed that the effect of FSH is consistent with that of AV-153 (DNA repair agent).

-The exact concentration of the compound that the samples are treated need to be listed together. They are listed in separate parts of the chapter, which makes them very confusing. The reader needs to return to the start of the paragraph to put altogether the different conditions on the samples.

Response: It was revised.

-The results need to be commented more. The difference between the treated samples with FSH and the untreated one can be easily distinguished by the graphs, but there is a lack of comments regarding these graphs. They are usually described as: Significant or effectively. These creates a confusion on the optimal amount of compound needed to obtain a positive result.

Response: It was revised.

The English language in general lacks clarity. There is a significant number of phrases that do not bring the point intended by the writer. Examples:

Response: It was revised.

Line[56-57]: This phrase is missing is between damage and caused.

Response: It was revised.

Line[59-60]:  Instead of don’t it should be used cannot.

Response: It was revised.

Line[89-90]: There is a in missing between cultured and 48-well culture plate.

Response: It was revised.

Line[93]: were instead of was

Response: It was revised.

Line[175-176]: This sentence is incorrect. The sentence should be: Meanwhile, the Golgi apparatus disintegrates in atresia follicles as observed by Transmission Electron Microscopy (TEM).

Response: It was revised.

Line[181]: off should be replaced with between.

Response: It was revised.

Line[205-206]: This sentence should be rephrased in order to highlight that the treatment with FSH shows a marked reduction in the expression of CHK2. The way this sentence is composed, is misleading and gives the illusion that after the treatment with ACNU it was obtained the wanted effect on CHK2.

Response: It was revised.

Figure 4E is not present in the Figure.

Response: It was revised.

Reviewer 3 Report

The manuscript was improved according to suggestions. However English needs more attention, microphotographs are too small. For immunoistochemistry provide concentrations of used antibodies not dillutions. There is no information on used FSH dose (was it based on dose-response analysis, literature data ?). Provide number of used animals and their age. What are exactly D580 hens or D280hens?-give more details

Author Response

Response to Reviewer

Re: cells-1623767

Title: Protective effect of follicle-stimulating hormone on DNA damage of chicken follicular granulosa cells by inhibiting CHK2/p53

We appreciate the review’s comments for improving the quality of our manuscript. The revisions were highlighted in yellow in the manuscript. The detailed responses to each question were listed as follows:

The manuscript was improved according to suggestions. However English needs more attention, microphotographs are too small.

Response: They were revised.

For immunoistochemistry provide concentrations of used antibodies not dillutions.

Response: The primary antibodies used for the immunofluorescence were as follows: rabbit anti-FSHR (1:200, GB11275-1, Servicebio, Wuhan, China), rabbit anti-γH2AX (1:200, A11463, Abclonal, Wuhan, China), rabbit anti-CDKN1A (1:200, ER1906-07), rabbit anti-CHK2 (1:200, ET1610-52, HUABIO, Hangzhou, China).

There is no information on used FSH dose (was it based on dose-response analysis, literature data ?).

Response: Treatment with 0.01 and 0.1 IU/ml FSH remarkably increased (37.55% and 52.00%) the cell viability.

Previous studies in our laboratory showed that the expression of apoptosis-associated genes caspase-3 was the lowest after treatment of 100 mIU/mL FSH. Meanwhile, the highest expression of Bcl2 and CDK2 proteins were achieved after FSH treatment at 100 mIU/mL in small white follicles.

Provide number of used animals and their age. What are exactly D580 hens or D280hens? -give more details

Response: It was revised. We used 280-day hens (laying peaking, more than 5). D580 hens was 580-day hens. D280 hens was 280-day hens.

Round 2

Reviewer 1 Report

The revised manuscript is partially improved, yet the authors did not address all the issues. The authors claimed that the antibodies, due to species limitations, did not work properly, I understand but they should find other tools/strategy to reinforce their data. 

Some part of the text should be improved in order to explain better the results (i.e. line 173, 174) through moderate English changes. Indeed, the authors added the IF staining of PUMA, but without any quantification.

Author Response

Response to Reviewer

Re: cells-1623767_R2

Title: Protective effect of follicle-stimulating hormone on DNA damage of chicken follicular granulosa cells by inhibiting CHK2/p53

We appreciate the reviewers’ comments for improving the quality of our manuscript. The revisions were highlighted in yellow in the manuscript. The detailed responses to each question were listed as follows:

The revised manuscript is partially improved, yet the authors did not address all the issues. The authors claimed that the antibodies, due to species limitations, did not work properly, I understand but they should find other tools/strategy to reinforce their data.

Response: We consolidated the data by RT-qPCR. In Figure1, we added the mRNA expression of CCND1, CHK2, CDKN1A and γH2AX to reinforce the data.

In Figure2, we added the mRNA expression of Rad51, CHK2, CDKN1A and γH2AX to reinforce the data.

In Figure5, we added the mRNA expression of Atg7, LC3B, MYO18A and GOLPH3 to reinforce the data.

Some parts of the text should be improved in order to explain better the results (i.e. line 173, 174) through moderate English changes.

Response: They were revised as “As shown in Fig. 1A, FSHR was predominantly expressed in the granulosa layer of both SWFs and ASWFs besides the vitelline membrane, but weakly expressed in the ASWFs.”.

Indeed, the authors added the IF staining of PUMA, but without any quantification.

Response: The quantification of PUMA was revised.

Furthermore, the immunofluorescence assay showed that compared with ACNU treat-ment, FSH treatment decreased the expression of PUMA by 24.78% (Fig. 3G).

Reviewer 2 Report

The manuscript has been improved after the revision. I suggest changing 'aging ovaries' instead of 'ovarian ageing' 

Author Response

Response to Reviewer

Re: cells-1623767_R2

Title: Protective effect of follicle-stimulating hormone on DNA damage of chicken follicular granulosa cells by inhibiting CHK2/p53

We appreciate the reviewers’ comments for improving the quality of our manuscript. The revisions were highlighted in yellow in the manuscript. The detailed responses to each question were listed as follows:

The manuscript has been improved after the revision. I suggest changing 'aging ovaries' instead of 'ovarian ageing'

Response: We use “ovary aging” to explain that it is the process of ovarian aging, not the aging ovary.

Reviewer 3 Report

Manuscript was corrected according suggestions.

Author Response

Response to Reviewer

Re: cells-1623767_R2

Title: Protective effect of follicle-stimulating hormone on DNA damage of chicken follicular granulosa cells by inhibiting CHK2/p53

We appreciate the reviewers’ comments for improving the quality of our manuscript. The revisions were highlighted in yellow in the manuscript. The detailed responses to each question were listed as follows:

Manuscript was corrected according suggestions.

Response: Thank you!

This manuscript is a resubmission of an earlier submission. The following is a list of the peer review reports and author responses from that submission.

Round 1

Reviewer 1 Report

The manuscript by Zhou et al. addresses an interesting subject relating to ageing in the chicken eggs. The authors seem to have performed extensive experimental work. However, I do not believe that the main hypotheses are verified experimentally and therefore I do not believe that this paper should be published by Cells.

One of the main hypotheses of the authors is that “damage of DNA was enhanced and the expression of FSH and FSHR were reduced in atretic follicles”. However, both these notions are not verified by Figure 1. In this Figure, it is not clear, from the immunofluorescence data, that FSHR is reduced in ASWFs. In addition, CDKN1A (p21) is a marker of a potential cell cycle checkpoint establishment leading to cell cycle arrest. It does not necessarily reflect DNA damage. A common marker for DNA damage (at least for double strand breaks and ICLs) is γH2AX. Therefore, throughout the whole manuscript, I do not believe that there is definite proof of DNA damage occurring.

The rise of p21 and Chk2 in ASWF may be showing that there is increased cell cycle arrest in ASWF. However, this possibility and the mechanisms in action, are not addressed by the authors.

The authors also state that “decreased DNA repaired occurred in the ASWF”. However, I did not find, in the manuscript, such a comparison between ASWF and SWF.

In relation to FSH, what may be happening is that FSH alleviates a state of cell cycle arrest (lower levels of p21 and maybe Chk2). However, we are not provided for an explanation for this, or the mechanisms involved.

Furthermore, I believe that some results are not presented correctly or accurately. For example:

- in Figure 1F, the p53 and actin bands seem saturated. How many WB were performed? There are no P values.

- in Figure S1, Cell viability with 0μM ACNU in (A) is the same as with 50μM ACNU in the absence of FSH in (B). That implies that in (B), ACNU has no effect. Furthermore, we see in the figure legend that AV-153 is used. However, there is no AV-153 in the actual figure.

- in Figure 2, the immunofluorescence experiment for Chk2 seems to give a different result compared to the one in the WBs.

- in Figure 7, why would Chk2 depletion cause more DNA damage and reduce DNA repair? What does it mean that Chk2 depletion “decreases cell cycle”? Chk2 is a DNA damage response checkpoint protein which becomes active in response to damage leading to cell cycle arrest. Why would its absence cause DNA damage?

Reviewer 2 Report

In the manuscript entitled “Protective effect of follicle-stimulation hormone on DNA damage of chicken follicular granulosa cells by inhibiting CHK2/p53”, the authors tested the protective effect of FSH towards the DNA damage response induced by aging in hens. The experiments were performed in vitro on granulosa cells (GCs), isolated from the two types of follicles, classified as small white follicles (SWFs) or small white follicles affected by atresia. The authors showed that FSH inhibited CHK2/p53 pathway, cell cycle arrest and autophagy in small follicles affected by atresia, following the treatment with ACNU.  

With major revisions, the manuscript could have its place in cells

Critical points:

Some compounds (i.e. “ACNU”, AV-153) are not defined in the text of the manuscript. The authors should describe in the text the rationale behind their choice to test such compounds in co-treatment with FSH in granulosa cells isolated from the follicles.

The reference at the end of this paragraph “Our previous studied have shown that FSH treatment efficiently reduced DNA damage and improve DNA repair via preventing the follicle atresia process in hens” is missing.

In the manuscript, the authors wrote that “GCs were grown in 6-wells plates at 90% confluency”. How many days are kept in culture before treatment? In the text, it is reported “A density of 4X105/well of cell was seeded in 6-wells culture plates (Nunc, Denmark) with chemicals”, the authors should describe in details the growth conditions for GCs, which chemicals they added in the medium? GCs are primary cells, derived from follicles, I imagine that in normal medium they will go in senescence very soon.

Western blot analysis of follicle, how many western blots were performed from independent experiments? It is not well defined how they did the quantification of the western blots. How many follicles were analyzed for each western blot?  In the western blot analysis, the authors should add a specific marker of germ cells (or granulosa cells), to indicate that have isolated an entire follicle (GCs+ oocyte).

Fig. 1 panel A. The staining was done on the entire follicle, GCs + oocyte in an ovarian section? The authors should add in the supplementary material the enlarged imagine of the entire follicles present in the ovarian section. The quality of IF image in Fig.1 panel B is very low, and western blot analysis of the same marker is not convincing (i.e. CHK2), or is missing (i.e. FSHR).

I have also some concerns, about the statement of the authors that effect of DNA damage on GCs via CHK2 activation. By IF assay, they detected the CHK2 expression, but not the CHK2 activation. To address this point they should use an antibody against phosphorylated CHK2, or a phosphorylated substrate of CHK2 (phosphorylated-p53).

Could the DNA damage response, that was induced by ACNU, be detected by gammaH2AX staining? If so, it will be nice to add such marker by IF.

Fig.2 panel C. Could the authors make a comment about the GCs morphology in presence of FSH? Although it is missing the staining with phalloidin, the morphology of GCs seems different, maybe the authors should add the co-staining with phalloidin in their IF analysis.

Fig.8 panel C.  It is not clear, how the authors performed the experiments, in tissue treated in culture with FSH? The sections indicate the small white follicle with atresia? Did they contain an oocyte or not?

Fig. 8 D, legend it is not clear how the GCs were treated with ACNU? (not indicated in the figure) with FSH? How long the GCs were treated? In addition, could the author make a comment about the upregulation of RAD51 or PCNA?  How many western blots from independent experiments were analyzed? The authors said that the expression of Caspase 3 was decreased in FSH-treated small white follicle with atresia. However, the data from western blot analysis Fig.8 panel D are not convincing. Maybe it is better to check for another apoptotic marker, cleaved Parp?

In conclusion, in the present form the manuscript cannot find its place in Cells, a more accurate presentation of data is required. The authors should also add in the supplementary material the original images of western blots